# COVID-19 Severity and Androgen Receptor Polymorphism

**DOI:** 10.3390/biology11070974

**Published:** 2022-06-28

**Authors:** Alessandra Buonacquisto, Anna Chiara Conflitti, Francesco Pallotti, Antonella Anzuini, Serena Bianchini, Luisa Caponecchia, Anna Carraro, Maria Rosa Ciardi, Fabiana Faja, Cristina Fiori, Daniele Gianfrilli, Andrea Lenzi, Miriam Lichtner, Claudio Maria Mastroianni, Patrizia Pasculli, Flavio Rizzo, Pietro Salacone, Annalisa Sebastianelli, Francesco Lombardo, Donatella Paoli

**Affiliations:** 1Laboratory of Seminology—Sperm Bank “Loredana Gandini”, Department of Experimental Medicine, “Sapienza” University of Rome, 00161 Rome, Italy; alessandra.buonacquisto@uniroma1.it (A.B.); annachiara.conflitti@uniroma1.it (A.C.C.); francesco.pallotti@uniroma1.it (F.P.); serena.bianchini@uniroma1.it (S.B.); fabiana.faja@uniroma1.it (F.F.); andrea.lenzi@uniroma1.it (A.L.); francesco.lombardo@uniroma1.it (F.L.); 2Hormone Laboratory, Department of Experimental Medicine, “Sapienza” University of Rome, 00161 Rome, Italy; antonella.anzuini@uniroma1.it; 3Andrology and Pathophysiology of Reproduction Unit, Santa Maria Goretti Hospital, 04100 Latina, Italy; luisa.caponecchia@gmail.com (L.C.); cristina.fiori82@gmail.com (C.F.); pietrosalacone@gmail.com (P.S.); annalisa.sebastianelli@gmail.com (A.S.); 4Infectious Diseases Unit, Santa Maria Goretti Hospital, Sapienza University of Rome, 04100 Latina, Italy; anna.carraro.92@gmail.com (A.C.); miriam.lichtner@uniroma1.it (M.L.); 5Department of Public Health and Infectious Diseases, “Sapienza” University of Rome, Policlinico Umberto I Hospital, 00161 Rome, Italy; maria.ciardi@uniroma1.it (M.R.C.); claudio.mastroianni@uniroma1.it (C.M.M.); patrizia.pasculli@uniroma1.it (P.P.); 6Section of Medical Pathophysiology and Endocrinology, Department of Experimental Medicine, Sapienza University of Rome, 00161 Rome, Italy; daniele.gianfrilli@uniroma1.it (D.G.); flavio.rizzo@uniroma1.it (F.R.)

**Keywords:** COVID-19, androgen receptor, polymorphism, testosterone

## Abstract

**Simple Summary:**

Since the early stages of the COVID-19 pandemic, it was recognized that male patients risked a more severe form of the disease. This suggested that, among other factors, a genetic predisposition could justify this issue. Thus, we investigated the CAG repeat polymorphism of the Androgen Receptor gene in a group of 142 recovered patients. Reduced androgenic signaling predisposes men to a more severe form: low testosterone levels and reduced androgen receptor activity (CAG > 23) expose the host to an excessive inflammatory response, leading downstream to the multi-organ damage seen in severe COVID-19.

**Abstract:**

During the COVID-19 pandemic, the most severe form of the disease was most often seen in male patients. The aim of this study was to identify any male predispositions that could be used to predict the outcome of the disease and enable early intervention. We investigated CAG polymorphism in the androgen receptor gene and serum levels of testosterone and LH, which were considered as probably responsible for this predisposition. The study involved 142 patients who had recovered from COVID-19 at least three months previously and were classified according to their disease severity using the World Health Organization (WHO) classification. We observed a significant increase in the number of CAG repeats with increasing disease severity: the percentage of patients with more than 23 repeats increased two-fold from Grade I to Grade IV. Furthermore, testosterone levels were significantly lower in patients with severe disease. Reduced androgenic signaling could predispose men to a more severe form: low testosterone levels and a reduced androgen receptor activity (CAG > 23) expose the host to an excessive inflammatory response, leading downstream to the multi-organ damage seen in severe COVID-19.

## 1. Introduction

Epidemiological data on SARS-CoV-2 infection highlight higher rates of hospitalization and mortality in men than in women. While an increased risk of death has been found in both sexes as age increases, above the age of 30, men have a significantly higher risk of death than women [1]. The WHO estimated that 63% of the COVID-19-related deaths in Europe were among men [2]. This disparity may be attributed to a number of factors, including immunological, hormonal and genetic differences. Immunological factors: women develop a strong, more robust, innate, cell-mediated and humoral immune response than men [3]. This difference is partly due to the sex chromosomes. The X chromosome has a high density of immune-related genes, some of which escape X chromosome inactivation: these genes are more expressed in women. For example, toll-like receptor 7 (TLR7), which is activated by estrogens, appears to play a crucial role in the early antiviral response, as it is involved in the detection of SARS-CoV-2 genetic material. Its higher expression in women partly explains how women are better able to overcome the early onset of a SARS-CoV-2 infection [3,4]. Hormonal factors: sex hormones are not only involved in the differentiation of reproductive organs, but they also exert sex-specific regulation on multiple tissues, including the brain and the immune system. Estrogens have been shown to have a protective role by regulating immune responses. In particular, by inhibiting the overproduction of pro-inflammatory cytokines, they prevent the cytokine storm involved in severe COVID-19. The protective function of estrogens is also associated with the activation of the ‘antiviral state’, mediated by type 1 interferon (IFN-1), the cytokine implicated in the forefront of host defense [5]. Conversely, androgens have different effects on both the innate and adaptive responses, often opposite to the effects of estrogens. Testosterone appears to play a key role in the pathogenesis of COVID-19. Its immunosuppressive role is, in fact, well-known, as it suppresses dendritic cells, thus reducing cytokine production and enhancing the production of immunosuppressive cytokines [6]. Genetic factors: androgen receptor regulation of the expression of transmembrane serine protease type II (TMPRSS2), a key factor for virus entry into host cells, is well-documented and is, to date, the only factor known to regulate the TMPRSS2 promoter [7]. In addition, literature data confirm an inverse correlation between serum testosterone levels and COVID-19′s severity [8,9,10,11,12]. As a whole, these data suggest testosterone has a dual role in the susceptibility to and clinical progression of COVID-19. The immunosuppressive action of testosterone and its regulation of TMPRSS2 gene transcription could explain why men are more susceptible to infection. Indeed, hypogonadism was found to be significantly correlated with the severity of COVID-19 [8,9,10,11,12]. Age-related low testosterone levels could lead to a loss of the immunosuppressive effect and, hence, to the cytokine storm typical of severe COVID-19 [8]. The effects of testosterone are mediated by the androgen receptor, a transcription factor in the nuclear receptor family, which is encoded by a gene on the X chromosome at Xq11–12 [13,14]. The AR gene has two polymorphic regions on exon 1, CAG and GGN, where the first includes CAG repeats and the second includes GGC repeats. The extreme variability of these repeats leads to the different lengths of the polyglutamine and polyglycine segments in the N-terminal transactivation domain [15]. The number of CAG repeats varies from 10 to 35, with an average in the Caucasian population of 21–22 [16,17]. A body of evidence suggests that the number of CAG repeats negatively correlates with AR transcriptional activity, both in vitro and in vivo [18,19]. As mentioned, the gene encoding for TMPRSS2 has been identified among the targets of AR activity; both in vitro and in vivo studies demonstrated that androgens regulate TMPRSS2 at both the transcriptional and post-translational levels [20]. While TMPRSS2 is most highly expressed in prostate tissue, it is also found in tissues targeted by SARS-CoV-2, such as the lungs [21]. From this perspective, a theoretical mechanism has been proposed to explain how polymorphism might be involved in the higher male mortality rate for COVID-19. It predicts that a low number of CAG repeats in the AR gene correlates with high receptor activity, which, by promoting the transcription of the TMPRSS2 gene, results in an increased risk of contracting a severe form of COVID-19. On the contrary, a high number of CAG repeats in the AR gene, correlated with a reduction in the receptor activity, results in a downstream-reduced TMPRSS2 transcription and, consequently, a low risk of severe COVID-19 disease [22]. However, there is still a lack of data, and the results of the only two studies presented in the literature appear contradictory [23,24].

Given the above, this study aimed to investigate:-Any correlation between the length of CAG polymorphism in the androgen receptor and COVID-19 disease severity;-Any correlation between serum LH and total testosterone levels and COVID-19 disease severity.

## 2. Materials and Methods

### 2.1. Patients

The study was approved by the “Sapienza” Ethics Committee (Ref. 0282/2021). The patients were recruited from the Infectious Disease Departments of Umberto I General Hospital—“Sapienza” and the University of Rome and Santa Maria Goretti Hospital, Latina. The patients contracted the infection during the first two waves of the pandemic, before the Italian vaccination campaign was opened to the whole population. Written informed consent was obtained from all the study participants. Patients were recruited according to the following inclusion criteria:-Nasopharyngeal swab, positive for SARS-CoV-2 between July 2020 and January 2021;-Age between 18–70 years;

The exclusion criteria were:-Andrological and/or systemic diseases that were capable of interfering with the gonadal hormone axis (autoimmune diseases, cardiovascular diseases, diabetes, etc.);-Genetic diseases;-Previous and/or current oncological disease and previous chemotherapy and/or radiotherapy treatments.

The caseload comprised 142 recovered patients whose COVID-19 severity was classified according to the WHO classification as Grade I: Mild (54 patients), Grade II: Moderate (38), Grade III: Severe, (30) or Grade IV: Critical (20) [25], and further stratified for statistical purposes into two subgroups: “severe/critical” (Grades III + IV) comprising 50 patients, and “mild/moderate” (Grades I + II) comprising 92 patients. Medical histories and other relevant clinical and biochemical data were retrieved from the patients’ medical records.

### 2.2. Genetic Analysis of Androgen Receptor Polymorphism

A molecular evaluation of the length of the polyglutamine segments of the AR gene was performed by fragment analysis, as described in Grassetti et al., 2014 [15]. A peripheral venous blood sample was taken from all patients. Genomic DNA was extracted from the peripheral blood leukocytes using the Wizard Genomic extraction kit (Promega Corporation, Madison, WI, USA). The concentration and purity were evaluated by a Nanodrop ND 1000 (Thermo Fisher Scientific, Waltham, MA, USA). The length of the polymorphic fragments (the number of CAG repeats) was analyzed by primers flanking the triplet repeat regions. Raw data from the capillary electrophoresis were analyzed by Gene Mapper Analysis (Applied Biosystems).

### 2.3. Hormone Evaluation

The peripheral blood sample was taken at around 8 a.m. after overnight fasting. The serum luteinizing hormone (LH) and total testosterone were quantified by the Chemiluminescent Microparticle Immuno Assay (CMIA, Architect System; Abbott Laboratories, Abbott Park, IL, USA). The detection limits, intra- and inter-assay coefficients of variation and normal ranges were as described in Pallotti et al., 2020 [26].

### 2.4. Statistical Analysis

The normality of the distributions of the continuous variables considered was assessed using the Kolmogorov–Smirnov test. The continuous variables were presented as mean ± standard deviations or a median and interquartile range, as most appropriate. The differences between groups were analyzed using the Mann–Whitney U or Kruskal–Wallis tests. For multiple comparisons, the results were adjusted post hoc, according to Bonferroni’s method. The categorical variables were then presented as counts and percentages and compared with Fisher’s χ² or exact tests. An evaluation of the statistically significant correlations among the examined variables was performed by Spearman’s rank correlation test. A *p*-value of less than 0.05 was considered statistically significant. Statistical analyses were performed with R software, version 4.0.2 (R Core Team (2020); R: A language and environment for statistical computing; R Foundation for Statistical Computing, Vienna, Austria. Available online: https://www.R-project.org/ (accessed date: 19 January 2022)).

## 3. Results

Age—The median age of Grades I and II patients was significantly lower than the Grade IV (Critical disease) group (*p* < 0.001; Figure 1A). The mean age of patients with “mild/moderate” disease was 40.8 ± 12.4, compared with 53.2 ± 10.7 for those with “severe/critical” disease (Figure 1B).

BMI—Anthropometric data were available for 91 subjects (mean BMI 27.0 ± 4.4). There was a significant correlation between the BMI and COVID-19 severity (Spearman’s ρ = 0.371; *p* < 0.001), with Grade I patients showing a significantly lower BMI than Grade IV patients (*p* = 0.010; Figure 2A). In general, the “mild/moderate” patients had a significantly lower BMI than the “severe/critical” cases (*p* = 0.037: Figure 2B).

Genetic analysis of androgen receptor polymorphism—The mean number of CAG repeats was 23.0 ± 2.9 (median 23.0). There was a statistically significant difference between the median length of polymorphism and disease severity (Grade I versus Grade IV, *p* = 0.050; “mild” versus “severe”, *p* = 0.016) (Table 1; Figure 3A,B). The percentage of subjects with more than 23 repeats doubled between Grade I and Grade IV (Figure 4), and there was a significant correlation between severe disease and >23 CAG repeats (Spearman’s ρ = 0.216; *p* = 0.010). The logistic regression models showed an OR of 3.56 for Grade I severity and ≤ 23 CAG repeats (Table 2).

Hormone evaluation—The median testosterone levels dropped as the disease severity increased, although the results were not significant (Table 1—Figure 5A,B). There was no statistically significant difference in the LH levels and LH/testosterone ratios between the four severity grades (Table 1). It should be noted that the hormone parameters of all the recruited subjects were within the reference limits for eugonadism (>12.0 nmol/L); therefore, the testosterone and LH levels were not suggestive of a hypogonadal condition.

## 4. Discussion

Androgen receptor polymorphism—The gender discrepancy in COVID-19 mortality and severity has a multifactorial etiology. Among other causes, genetic factors, specifically the androgen receptor’s genetic variations, can contribute to disease severity [27]. As is well-known, proteolytic cleavage of the viral spike protein by transmembrane serine protease 2 (TMPRSS2) is required for SARS-CoV-2 to enter host cells, and the androgen receptor regulates TMPRSS2 gene transcription. AR transcriptional activity is associated with the length of the tandem CAG repeats of exon 1 [14], and a large body of evidence suggests that the number of CAG repeats is negatively associated with AR transcriptional activity, both in vitro and in vivo [18,19]. Mohamed et al. hypothesized a theoretical mechanism in which the polymorphic CAG repeat length is correlated with COVID-19 severity [22]. This hypothesis is supported by the observation that the CAG repeat length seems to underlie the observed ethnic differences in COVID-19 mortality rates [27]. However, the only papers on this topic deviate from the proposed theoretical mechanism. Baldassari et al., comparing asymptomatic patients and patients with severe COVID-19, noticed that most patients with fewer than 22 CAG repeats were asymptomatic. The authors hypothesized that the low triplet number is associated with a more active AR, which may mediate the immunosuppressive effects of testosterone, thus counteracting the cytokine storm typical of the severe form of COVID-19 [23]. In line with these findings, McCoy et al. showed that COVID-19 patients with fewer than 22 CAG repeats had a lower risk of admission to an intensive care unit (ICU) and were hospitalized for significantly fewer days than patients with more than 23 repeats [24].

The genetic analysis performed in this study gave results that were consistent with those of Baldassari et al. and McCoy et al. [23,24] (Table 3). By stratifying the patients into the four disease severity grades, we observed a significant increase in the number of CAG triplets as severity increased. Specifically, comparing Grade I with Grade IV, the percentage of patients with more than 23 repeats doubled, which is associated with an increased risk (>3 fold) of severe disease. Given the inverse relationship between the number of triplets and androgen receptor activity, these results could allow us to hypothesize that a longer polymorphic tract increases the risk of developing a severe form of COVID-19 due to a reduction in receptor-mediated immunosuppressive signaling. In the presence of a less active androgen receptor, the immunosuppressive effects of testosterone are not fully mediated, exposing the body to the cytokine storm responsible for the multi-organ damage seen in COVID-19.

Hormone Profile: LH and Testosterone in COVID-19—The high male mortality rate observed during the COVID-19 pandemic, and the role of testosterone in modulating the immune response prompted an investigation into the possible correlations between the serum levels of the hormone and the disease’s severity. It has been hypothesized that an individual’s hormonal environment may play an important role in both the susceptibility to infection and the severity of the disease’s clinical course [8]. Although the various studies agree that low serum testosterone is correlated with disease severity, its role in the pathogenesis of COVID-19 is much more complicated [8,9,10,11,12]. Salciccia et al. noted an inverse correlation between testosterone levels and IL-6 levels, a key factor in the cytokine storm that is characteristic of severe COVID-19. The authors hypothesized that higher testosterone levels could act as a hormonal shield against the COVID-19-related cytokine syndrome [8]. Consistently, with these findings, Camici et al. found an inverse correlation between testosterone levels and COVID-19′s hyperinflammatory syndrome markers, suggesting that a transient state of primary hypogonadism may develop as a consequence of direct damage to the testis epithelium by SARS-CoV-2 [9]. Similarly, Rastrelli et al. highlighted that lower basal total testosterone levels predicted a poor prognosis and increased mortality in 31 men with SARS-CoV-2 pneumonia [12]. An inverse correlation between the serum testosterone levels and clinical progression in patients with COVID-19 was also observed by Cinislioglu et al. In their study, serum testosterone levels were significantly lower in patients with a severe form of COVID-19 compared with those presenting with a milder form; the same difference was also observed when comparing patients who required intensive care and patients who did not [11]. Finally, Salonia et al. stratified patients with COVID-19 according to different types of hypogonadism: total testosterone and LH levels were suggestive of secondary hypogonadism in 85% of the patients analyzed [10]. These observations confirm that hypogonadism is a risk factor for a worsening clinical progression and also suggest that this condition and the resulting loss of immunosuppressive effects may represent a pathogenic viral mechanism targeting the testicular tissue. Nonetheless, the lack of testicular co-expression of ACE2 and TMPRSS2 precludes the direct involvement of this tissue; consequently, hypogonadism could be an indirect mechanism by which the virus alters the central regulation of gonadal function through the effectors of the hyperinflammatory syndrome [28].

Hypogonadism is more prevalent in older men with obesity and comorbidities, such as dysmetabolic disorders—precisely those who have presented the most critical form of COVID-19 [8,29,30]. Thus, based on the role of androgens in the immune response and the change in androgen levels throughout life, it can be hypothesized that testosterone may be a double-edged sword in the natural history of COVID-19 infection. In the early phase, its immunosuppressive action might explain the greater susceptibility of men than women of all ages to infection. Once an infection has been established in older men, who also frequently develop acute respiratory distress syndrome (ARDS), lower age-related testosterone levels could result in a lower immunosuppressive effect and a more robust cytokine response. Hypogonadism may, therefore, play a protective role in early COVID-19 infections; however, it may also lead to a more severe clinical course [8]. The evaluation of the hormone profile in this study revealed results that were in line with the numerous studies in the literature [8,9,10,11,12]. When comparing the four severity classes, we observed a negative trend in testosterone levels in correlation with disease severity, while in comparing the “severe/critical” and “mild/moderate” groups, the testosterone levels in the former were significantly lower. Although the testosterone levels were reduced, they were not indicative of hypogonadism. In light of these data and the role of testosterone in the immune response, it can be assumed that low testosterone levels are unable to start immunosuppressive pathways. This could result in an imbalance between the immunosuppressive and pro-inflammatory regulatory mechanisms, which are in favor of the latter. This event could cause an excessive increase in the pro-inflammatory cytokine-mediated response following a SARS-CoV-2 infection, known as a cytokine storm. The final result is a severe injury to host organs and tissues and an increased risk of developing a severe form of the disease.

Taken collectively, these results have interesting repercussions on subjects undergoing androgen suppression treatments. The immunological imbalance complicated by the testosterone blockade could represent an additional significant risk factor for a severe-to-critical disease in these subjects since the treatment reduces the androgen signaling and, likely, its anti-inflammatory function. While the literature data show that androgen deprivation therapy does not modify the risk of the infection itself [31], data on its influence on the prognosis are controversial. In fact, a recent trial failed to detect significant differences between those subjects undergoing anti-androgenic treatment and the controls [32], and a metanalysis of the available data could only show a trend of risk reduction [33]. Thus, the available data still do not support the in vivo protective role of androgens, and more studies are needed to elucidate this effect.

Limits and strengths—We presented several interesting results in a substantial caseload; nonetheless, since we investigated the role of gene polymorphism, a further increase in the caseload is necessary to confirm our results. On the other hand, we recruited subjects among those performing a follow-up visit in an infectious diseases department, and as such, they are more likely to represent the general population.

## 5. Conclusions

In conclusion, our study showed that reduced androgenic signaling might predispose men to a severe form of COVID-19. These results further confirm the immunosuppressive role of testosterone, as low levels and a reduced (CAG > 23) activity of the testosterone receptor expose the host to an excessive inflammatory response, leading downstream to the multi-organ damage typical of a severe form of COVID-19. Our data also allows us to answer several questions regarding the evident difference in the clinical course of the disease in the two sexes, identifying a genetic cause responsible for the male predisposition to severe COVID-19.

## Figures and Tables

**Figure 1 biology-11-00974-f001:**
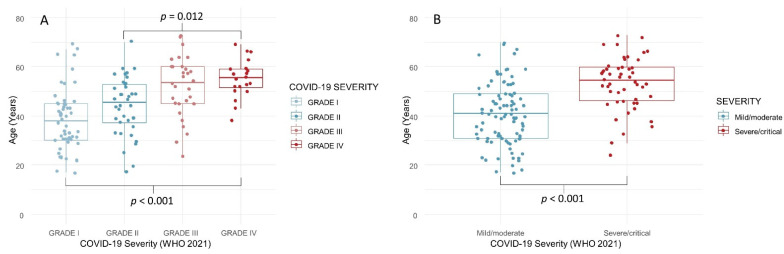
Age of COVID-19 survivors, according to disease severity (4 grades, (**A**); 2 grades, (**B**)).

**Figure 2 biology-11-00974-f002:**
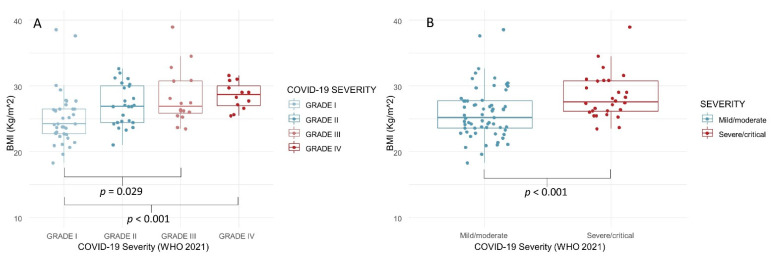
BMI of COVID-19 survivors, according to disease severity (4 grades, (**A**); 2 grades, (**B**)). BMI (body mass index).

**Figure 3 biology-11-00974-f003:**
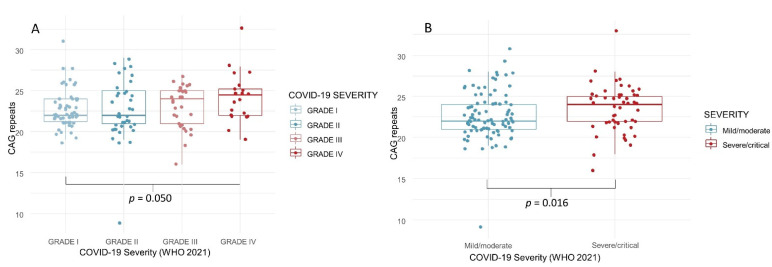
Number of CAG repeats in COVID-19 survivors, according to disease severity (4 grades, (**A**); 2 grades, (**B**)).

**Figure 4 biology-11-00974-f004:**
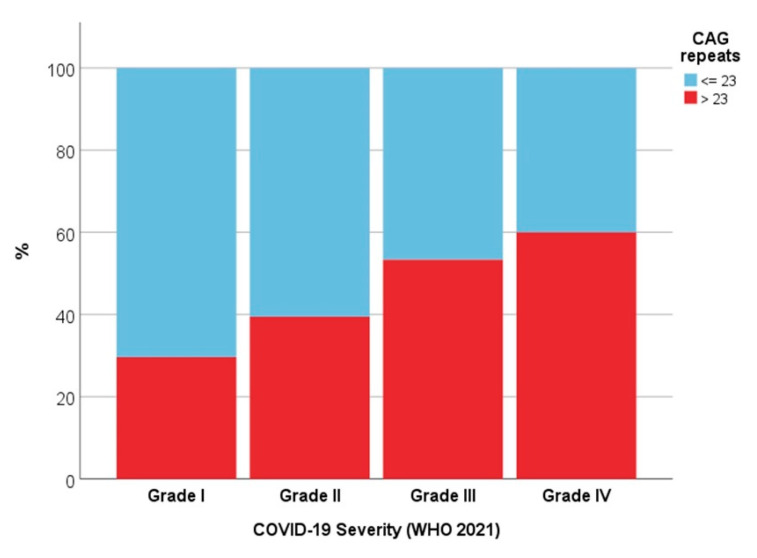
Percentages of subjects with ≤23 or >23 CAG repeats, according to disease severity (4 grades).

**Figure 5 biology-11-00974-f005:**
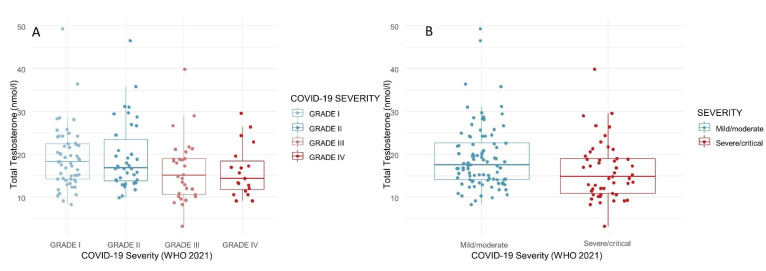
Total testosterone (nmol/L) in COVID-19 survivors, according to disease severity (4 grades, (**A**); 2 grades, (**B**)).

**Table 1 biology-11-00974-t001:** Mean ± standard deviations, median (in italics) and 25th–75th percentile (in brackets) of relevant variables stratified by COVID-19 severity.

	Age (Years)	BMI (Kg/m^2^)	CAG Repeats	LH (mIU/mL)	Total Testosterone (nmol/L)	LH/TT Ratio
**Grade I**	38.4 ± 12.5	25.1 ± 4.2	22.8 ± 2.3	4.1 ± 1.9	19.0 ± 7.1	0.25 ± 0.14
	*38.0*	*24.2*	*22.0*	*4.0*	*18.2*	*0.22*
	(30.0–45.0)	(22.7–26.5)	(21.0–24.0)	(3.0–5.1)	(14.1–22.5)	(0.14–0.33)
**Grade II**	44.2 ± 11.7	27.1 ± 3.1	22.6 ± 3.6	3.4 ± 3.2	19.5 ± 8.4	0.18 ± 0.11
	*45.5*	*26.9*	*22.0*	*3.0*	*16.8*	*0.16*
	(37.0–53.0)	(24.4–30.0)	(21.0–25.0)	(1.9–3.9)	(13.8–24.4)	(0.10–0.19)
**Grade III**	52.0 ± 12.3	28.3 ± 4.2	23.0 ± 2.7	4.5 ± 3.4	16.1 ± 7.3	0.32 ± 0.25
	*53.5*	*26.9*	*24.0*	*3.5*	*15.0*	*0.24*
	(45.0–60.0)	(25.7–30.8)	(21.0–25.0)	(2.6–5.4)	(10.5–19.0)	(0.16–0.39)
**Grade IV**	55.1 ± 7.8	30.3 ± 5.0	24.3 ± 3.1	3.9 ± 2.6	16.1 ± 6.0	0.26 ± 0.14
	*55.0*	*29.0*	*24.5*	*3.1*	*14.3*	*0.15*
	(51.0–59.0)	(27.2–31.0)	(22.0–25.5)	(2.3–4.6)	(11.4–19.6)	(0.21–0.34)
***p*-value**	<0.001	0.002	0.050	0.055	0.435	0.156

**Table 2 biology-11-00974-t002:** Results of logistic regression between COVID-19 severity (dependent variable) and the number of CAG repeats (independent variable). * Grade IV is considered as a reference. Key: OR: Odds Ratio, CI = confidence interval.

		OR	95% CI	*p* Value
**Grade I**	CAG repeats ≤ 23	3.56	1.22–10.37	0.020
**Grade II**	CAG repeats ≤ 23	2.30	0.76–6.95	0.140
**Grade III**	CAG repeats ≤ 23	1.31	0.42–4.13	0.642
**Grade IV ***	CAG repeats ≤ 23	//	//	//

**Table 3 biology-11-00974-t003:** Comparison of the results of this study with the results of published papers.

Papers	COVID-19 Severity	N. AR CAG Repeats
**McCoy et al., 2021**	Severe	≥22
**Baldassarri et al., 2021**	Severe	≥23
**Present study, 2022**	Severe	>23

## Data Availability

Data included in this paper can be made available upon reasonable request to the corresponding author.

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
