# Peer review of "COVID-19 Severity and Androgen Receptor Polymorphism"

_biology, 2022, doi:10.3390/biology11070974_

Round 1

Reviewer 1 Report

GENERAL COMMENT

The authors investigated the role of the CAG polymorphism as a possible responsible for the preparation of men to the more severe form of COVID-19 with a cross-sectional study on 142 patients.

Some major issues are present together with several minor issues.

MAJOR CORRECTIONS

I suggest to soften the conclusion in both abstract and main-text. The limitations of study are significant for strong conclusions.

The study was approved by the “Sapienza” Ethics Committee (Ref. 0282/2021). The ID is referred to 2021, therefore, theoretically, enrollment should have started in 2021. However, the study period is July 2020-January 2021. The authors should clarify this relevant concern.

Line 118: “Nasopharyngeal swab positive for SARS-CoV-2”. Were PCR or antigenic test performed? The authors should specify this point

Line 122: The authors should specify what “systemic diseases” means.

Line 130 (and other parts of text, figures and tables): I suggest to replace “mild” in “mild-to-moderate”

I suggest a brief section with the limitations of study

MINOR CORRECTIONS

Lines 28 and 41: I suggest to specify “androgen” receptor

I suggest to cite this interesting paper on COVID-19 https://pubmed.ncbi.nlm.nih.gov/34017152/

Figures: the authors should replace “,” with “.” to separate the decimals. Besides, the meanings of all acronyms should be specified in the legend of each figure

Author Response

Reviewer 1

GENERAL COMMENT

The authors investigated the role of the CAG polymorphism as a possible responsible for the preparation of men to the more severe form of COVID-19 with a cross-sectional study on 142 patients. Some major issues are present together with several minor issues.

MAJOR CORRECTIONS

Q1: I suggest to soften the conclusion in both abstract and main-text. The limitations of study are significant for strong conclusions.

A1: We thank the reviewer for his suggestion. We softened the conclusion, and we added a brief section with the limitation of study.

Q2: The study was approved by the “Sapienza” Ethics Committee (Ref. 0282/2021). The ID is referred to 2021, therefore, theoretically, enrollment should have started in 2021. However, the study period is July 2020-January 2021. The authors should clarify this relevant concern.

A2: We thank the reviewer for the possibility to clarify this point. We enrolled these patients “at least three months from recovery” (the first negative PCR test from a pharyngeal swab) during their follow up visits in the two participating Infectious Diseases departments. The enrolled patients actually had a “nasopharyngeal swab positive for SARS-CoV-2 between July 2020 and January 2021”. The first EC approval for the Rome center only is dated 14th September 2020 (Ref. 0646/2020). To increase the number of patients recruited, an amendment was requested and approved (the latest version of the EC approval is the one referred in the manuscript – 2nd April 2021 - Ref. 0282/2021) in order to include the Latina center. In both cases, in each center we started collecting blood samples for androgen receptor sequencing (and other study-related procedures) only after the corresponding EC approval.

Q3: Line 118: “Nasopharyngeal swab positive for SARS-CoV-2”. Were PCR or antigenic test performed? The authors should specify this point

A3: All diagnoses were based on RT-PCR. This has been clarified in the text.

Q4: Line 122: The authors should specify what “systemic diseases” means.

A4: We clarified in the manuscript this point.

Q5: Line 130 (and other parts of text, figures and tables): I suggest to replace “mild” in “mild-to-moderate”

A5: We agree with the reviewer. We modified this issue within the text.

Q6: I suggest a brief section with the limitations of study

A6:  We thank the reviewer for this suggestion and we added a brief section with the limitation of our study within the text. “

MINOR CORRECTIONS

Q7: Lines 28 and 41: I suggest to specify “androgen” receptor

Q8: I suggest to cite this interesting paper on COVID-19 https://pubmed.ncbi.nlm.nih.gov/34017152/

Q9: Figures: the authors should replace “,” with “.” to separate the decimals. Besides, the meanings of all acronyms should be specified in the legend of each figure

A7-9: The text has been modified as requested.

Reviewer 2 Report

Interesting paper showing that the length of AR polyQ-repeats (CAG>23) as well as low levels of circulating testosterone support a more severe form of Covid-19. The paper is well written, data and conclusions are sound.

Suggestions:

As reduced AR-signalling predisposes men to a more severe from of Covid-19 the authors shoud discuss the implications of their findings for patients receiving androgen synthesis inhibitors and or/antiandrogens.

Author Response

Reviewer 2

Interesting paper showing that the length of AR polyQ-repeats (CAG>23) as well as low levels of circulating testosterone support a more severe form of Covid-19. The paper is well written, data and conclusions are sound.

Suggestions:

Q1: As reduced AR-signalling predisposes men to a more severe from of Covid-19 the authors should discuss the implications of their findings for patients receiving androgen synthesis inhibitors and or/antiandrogens.

A1: We thank the reviewer for his relevant suggestion. We added a short paragraph as requested.

Round 2

Reviewer 1 Report

The authors changed the paper according to my suggestions.